# Deubiquitinating Enzymes Ubiquitin-Specific Proteases 7 and 10 Regulate TAU Aggregation

**DOI:** 10.3390/ijms262211062

**Published:** 2025-11-15

**Authors:** Christiane Volbracht, Karina Fog

**Affiliations:** Neuroscience, H. Lundbeck A/S, 2500 Valby, Denmark

**Keywords:** Alzheimer’s disease, tauopathy, rTg4510, ubiquitin, USP7, USP10

## Abstract

Accumulation of the microtubule-associated protein TAU into inclusions is a hallmark of tauopathies including Alzheimer’s disease (AD), potentially driven by impaired protein degradation and dysregulated ubiquitination. To explore the role of deubiquitinating enzymes (DUBs), we performed siRNA knockdown screens targeting 93 murine DUBs in rTg4510 cortical cultures. Knockdown and pharmacological inhibition of the ubiquitin-specific proteases 7 (Usp7) and 10 (Usp10) significantly reduced seeded TAU aggregation without affecting soluble TAU levels. These effects were observed in both cortical and organotypic hippocampal slice cultures from rTg4510 mice, as well as in wildtype neurons seeded with AD-derived pathological TAU. Inhibition of Usp7 and Usp10 was associated with increased polyubiquitination of residual TAU inclusions in rTg4510 cortical cultures. These findings suggest that Usp7 and Usp10 contribute to pathological TAU accumulation by modulating ubiquitin-dependent degradation pathways. Targeting USP7 and USP10 may offer a novel therapeutic strategy for AD and related tauopathies.

## 1. Introduction

Intracellular accumulation of the microtubule-associated protein TAU (MAPT) as filamentous aggregates is a hallmark of tauopathies, including Alzheimer’s disease (AD) [1]. These aggregates, mainly composed of hyperphosphorylated TAU in the form of paired helical filaments (PHFs), form neurofibrillary tangles (NFTs) [2], whose distribution correlates with cognitive decline and disease progression in AD [3,4]. Mutations in the *MAPT* gene linked to familial tauopathies [5,6] accelerate TAU hyperphosphorylation, oligomerization, and aggregation [7,8]. TAU aggregation follows a nucleation-dependent process [9], which can be accelerated by pathological TAU “seeds” that template normal TAU and amplify TAU pathology in vivo [10] and in cultured neurons recapitulating some features of tauopathies in vitro [11,12,13].

TAU is primarily degraded by two systems: the ubiquitin–proteasome system (UPS) and the autophagy–lysosomal pathway [14]. In the UPS, target proteins are tagged for degradation via ubiquitination by concert activity of a ubiquitin-activating enzyme E1, a ubiquitin-conjugating enzyme E2, and a ubiquitin E3 ligase and processed by the 26S proteasome. Soluble TAU oligomers and filamentous TAU aggregates can be cleared by this pathway [15,16]. Approximately 100 deubiquitinases (DUBs), which remove ubiquitin from target proteins, counteract E3 ligase activity preventing the degradation of target proteins [17]. Most DUBs are cysteine proteases and belong to the families of ubiquitin-specific proteases (USPs), sentrin-/SUMO-specific protease (SENPs), ovarian tumor proteases (OTUs), Machado–Josephin domain proteases (MJDs), ubiquitin carboxyl-terminal hydrolases (UCHs), and motif-interacting with ubiquitin-containing novel DUB family proteases (MINDYs). The family of JAB1/MPN/MOV34 proteases (JAMMs) are zinc metalloproteases [17].

Dysregulation of TAU ubiquitination, through altered E3 ligases or DUBs activity, is suggested to contribute to TAU accumulation in AD [14]. Pathological TAU is ubiquitinated [18,19] mainly by the E3 ligase carboxyl terminus of the Hsc70-interacting protein (CHIP) [20,21], which colocalizes with NFTs [20] and is inversely correlated to PHF-TAU levels in AD brains [22]. Upregulation of TAU ubiquitination promotes clearance of pathological TAU as overexpression of CHIP enhances degradation [23,24] while deletion enhances accumulation of hyperphosphorylated TAU in rodent brains [25]. Several DUBs such as USP10 [26], USP11 [27], USP13 [28], and UCH-L1 [29] are upregulated in AD brains and USP10 [30] and UCH-L1 [31] colocalize with TAU lesions. Respective DUBs such as OTU domain-containing ubiquitin aldehyde binding 1 (OTUB1) [32], UCH-L1 [33], USP7 [34], USP9 [35], USP10 [26], USP11 [27], USP13 [28], and USP14 [36,37] can remove ubiquitin from TAU and are considered attractive targets to enhance TAU ubiquitination and thereby degradation [38].

We hypothesized that DUB activity contributes to the accumulation of pathological TAU and investigated whether silencing individual DUBs could influence seeded TAU aggregation in neuronal cultures. Through this approach, we identified Usp7 and Usp10 as key regulators. Both genetic and pharmacological inhibition of these two DUBs significantly reduced seeded TAU aggregation across multiple neuronal models, without altering soluble TAU levels. This reduction was accompanied by increased polyubiquitination of the remaining TAU inclusions, suggesting that Usp7 and Usp10 activity regulates ubiquitination of pathological TAU and thereby its degradation.

## 2. Results

### 2.1. Defining Optimal Conditions for Seeded TAU Aggregation in Cortical Cultures (CTX) from rTg4510 Mice

To establish optimal conditions for seeded TAU aggregation in neuronal cultures, we first performed concentration-finding experiments. We induced seeded TAU aggregation in cortical cultures (CTX) isolated from rTg4510 embryos using either S1p or P3 fractions extracted from rTg4510 mouse brains containing oligomeric and fibrillar hyperphosphorylated TAU species, respectively, as established previously [39,40]. CTX overexpress the 4R0N isoform of human TAU with the P301L mutation in addition to endogenous murine Tau and exhibit no spontaneous TAU hyperphosphorylation and aggregation when not seeded (Figure 1A, Appendix A). S1p and P3 TAU seed uptake into CTX occurred unassisted within 24 h as we reported previously [41] and induced seeded TAU aggregation characterized by AT8-positive (Appendix A) and pS422-TAU-positive inclusions which co-stained for polyubiquitin detected with the anti-ubiquitin antibody P4D1, decorating the neuronal bodies and networks (Figure 1A). We confirmed that these inclusions contained hyperphosphorylated human 4R0N TAU species (64 kDa) which were exclusively detected in the insoluble SDS fraction isolated from seeded CTX by Western blotting (Appendix A). When CTX were incubated at DIV 7 with increasing concentrations (0.2 to 8 ng/uL) of oligomeric S1p or fibrillary P3 TAU seeds isolated from brains of 32–40 weeks old rTg4510 mice, we observed concentration-dependent seeded TAU aggregation at DIV 15 quantified by the human TAU aggregation assay (Figure 1B). To ensure that brains from 32 to 40 weeks old rTg4510 mice were optimal for pathological TAU seed isolation, we incubated CTX at DIV 7 with 2 ng/uL S1p or P3 TAU seeds isolated from brains of rTg4510 mice from different ages (8, 16, 24, 32, 40, 48, and 56 weeks old) and of 56 weeks old non-transgenic (non-tg) mice as controls. Quantification of seeded TAU aggregation at DIV 15 showed increased seed potency up to 32 weeks age whereafter it plateaued (Figure 1C), consistent with the reported increasing TAU pathology observed in aged rTg4510 mice [42]. For subsequent experiments, CTX were incubated with 2 ng/uL S1p or P3 TAU seeds from brains of 32–40 weeks old rTg4510 mice at DIV 7 and seeded TAU aggregation investigated at DIV 15. Seeded TAU aggregation did not result in impaired viability measured by resazurin assay and accessed by healthy neuronal networks and nuclei (Figure 1A and Appendix A). Equivalent seeded TAU aggregation was induced with S1p and P3 seeds (Figure 1A–C), and due to feasibility, the former was selected to induce seeded TAU aggregation in the subsequent DUB knockdown screens, whereas the P3 seeds were included in all subsequent hit validations.

We next investigated whether this model could be applied for genetic knockdown screening on seeded TAU aggregation. To achieve efficient gene knockdown in CTX from rTg4510 mice, we selected the transfection-free Accell siRNA and first knocked down human *MAPT*, which encodes TAU protein. Treatment of CTX with MAPT siRNA #13 (siMAPT) at DIV 1, DIV 4, and DIV 6 (before seed application at DIV 7) led to efficient knockdown of human TAU protein and nearly complete inhibition of seeded TAU aggregation at DIV 15 (Figure 1D). When siMAPT was applied at DIV 8 and DIV 10 (after seed application at DIV 7), approximately 75% and 25% TAU knockdown and corresponding inhibition of seeded TAU aggregation were achieved, respectively, at DIV 15 (Figure 1D). We observed a strong correlation between the knockdown of human TAU protein and prevention of seeded TAU aggregation in CTX from rTg4510 mice. Based on these findings, we selected siRNA application at DIV 1, DIV 4, and DIV 6 for the subsequent DUB knockdown screens with the rational of employing different times to initiate silencing to circumvent possible compensation and toxic events which we speculated could occur by prolonged DUB depletion. However, most DUB siRNA hits were identified by knockdown initiated at DIV 1 and DIV 4 and when siRNAs impaired neuronal viability, toxicity was mostly observed equally in all three siRNA addition schemes (see below). All optimized parameters for the screens are summarized in Figure 2A. Lastly, we investigated whether enhancing cellular degradation by inducing autophagy could capture changes in seeded TAU aggregation in CTX from rTg4510 mice. The autophagy inducers trehalose and rapamycin were incubated at DIV 8 in the otherwise unchanged screening setup and decreased seeded TAU aggregation in a concentration-dependent manner (Appendix A), validating that the defined parameters of the screen allowed the identification of proteostasis modifiers of seeded TAU aggregation.

### 2.2. Focused DUB siRNA Screen on Seeded TAU Aggregation in CTX from rTg4510 Mice

In preparation for the Accell siRNA DUB knockdown screen, we determined intra and inter plate variations in seeded TAU aggregation induced by S1p TAU seeds in the plate layout to be used for the siRNA screen. CTX were plated in the middle 60 wells of three 96-well plates and incubated with S1p seeds at DIV 7 for 24 h followed by assessment of cellular viability with the resazurin assay and seeded TAU aggregation with the human TAU aggregation immunoassay at DIV 15. We determined acceptable 3–16% intra and 6–8% inter plate variations for S1p-induced seeded TAU aggregation (Appendix A) and 3–15% intra and 8–10% inter plate variations for viability on these three test plates. We next evaluated the robustness of the signal-to-noise ratio of our screening assay by measuring the Z’ value. Robust assays display Z’ values of >0.5 showing a statistical difference of 12 SDs between the positive and negative controls [43]. Our positive and negative controls were seeded CTX treated with siMAPT in technical quintuplicates and non-targeting smartpool siRNA in technical decemplicates, respectively. We calculated Z’ values of 0.66, 0.67, and 0.56 with siRNA incubation at DIV 1, DIV 4, and DIV 6, respectively, in seeded CTX (Appendix A) and concluded that the overall signal-to-noise ratio in the human TAU aggregation immunoassay was sufficient for screening.

A focused Accell smartpool siRNA knockdown screen against 93 murine DUBs (Appendix A) was employed to investigate the effect on seeded TAU aggregation in CTX. Smartpool siRNA contains a mixture of four individual siRNA sequences to target the same gene, providing a more guaranteed gene knockdown than using a single siRNA sequence. A total of 1 µM smartpool siRNA was added either at DIV 1, DIV 4, or DIV 6, followed by 2 ng/µL S1p TAU seeds at DIV 7. Viability by resazurin assay and seeded TAU aggregation by the human TAU aggregation immunoassay were determined at DIV 15. Each screening plate contained, in technical quintuplicates, the respective smartpool DUB siRNAs, the positive control siMAPT, and the siRNA dilution buffer, and in technical decemplicates the negative control non-targeting smartpool siRNA (Appendix A). We calculated Z’ values based on the negative control non-targeting smartpool siRNA in technical decemplicates and the positive control siMAPT in technical quintuplicates for each screening plate and excluded plates with Z’ values < 0.4 from analysis, which were rerun in new experiments.

Based on the measured plate variations, we excluded smartpool siRNAs affecting viability by more than 20% as toxic and defined primary hits as smartpool siRNAs leading to more than 20% change in seeded TAU aggregation independent of statistical significance. We identified 23, 29, and 16 DUB siRNAs modulating seeded TAU aggregation without inducing toxicity with DIV 1 (Appendix A), DIV 4 (Figure 2A), and DIV 6 (Appendix A) addition, respectively. Primary siRNA hits were nearly completely matching in the three application schemes, as
most of the 23 hits from the DIV 1 addition and of the 16 hits from the DIV 6 addition were included in the 29 hits identified from the DIV 4 addition. The majority of hits affected seeded TAU aggregation alike either decreasing or increasing across the three application schemes. From the 29 DUB siRNA hits modulating seeded TAU aggregation, 7 increased and 22 decreased seeded TAU aggregation and from the latter, 10 significantly reduced seeded TAU aggregation with siRNA application at DIV 4 (Figure 2A). None of the 7 DUB siRNAs which increased seeded TAU aggregation reached significance, and many displayed 10–20% lower viability measurements (Figure 2A). Smartpool siRNA Usp42 led to 30% toxicity in all three application schemes and was excluded.

#### 2.2.1. Confirmation of Primary Hits in Seeded TAU Aggregation Induced with S1p TAU Seeds

Smartpool siRNAs from total 32 primary DUB hits were reordered and rescreened for effects in S1p-induced seeded TAU aggregation with addition schemes at DIV 1, DIV 4, and DIV 6. In these rescreens we confirmed 11 siRNA DUB hits (Usp4, Usp7, Usp9x, Usp10, Usp21, Usp30, Usp33, Otud7a, Eif3h, Senp3 and Senp5) which reduced S1p-induced seeded TAU aggregation > 20% in the DIV 4 application scheme (Figure 2B). Five DUB siRNAs (Usp7, Usp9x, Usp10, Usp21, and Senp3) significantly decreased seeded TAU aggregation when added at DIV 4 (Figure 2B). None of the 8 DUB siRNA hits increasing seeded TAU aggregation could be confirmed in any of the three addition schemes (Figure 2B, Appendix A). With siRNA application at DIV 1, we confirmed 8 siRNA DUB hits (Usp7, Usp9x, Usp10, Usp21, Otud7a, Eif3h, Senp3 and Senp5) lowering S1p-induced seeded TAU aggregation > 20%, of which Usp7, Usp10, Usp21, Otub7a, Eif3h, and Senp3 siRNA reached significance (Appendix A). In DIV 6 scheme, 9 DUB siRNA hits (Usp7, Usp9x, Usp10, Usp21, Usp30, Usp33, Otub7a, Eif3h, and Senp3) were confirmed, reducing S1p-induced seeded TAU aggregation > 20%, with Usp7, Usp9x, Usp10, Otub7a, Eif3h, and Senp3 siRNA significantly decreasing seeded TAU aggregation (Appendix A). We continued subsequent validation studies with siRNA addition at DIV 1 and DIV 4, in which most hits were identified.

#### 2.2.2. Validation of Hits in Seeded TAU Aggregation Induced with P3 TAU Seeds

Smartpool siRNAs for the 11 confirmed DUB hits were reordered and validated in both S1p- and P3-induced seeded TAU aggregation. To this end, in parallel experiments, 1 µM smartpool siRNA was added at DIV 1 or DIV 4, 2 ng/µL S1p or P3 TAU seeds were added at DIV 7 and seeded TAU aggregation was measured by the human TAU aggregation immunoassay at DIV 15. We confirmed 5 siRNA DUB hits, Usp7, Usp9x, Usp10, Usp21 and Senp3, which led to >20% reductions in S1p- and P3-induced seeded TAU aggregation with their respective smartpool siRNA in DIV 1 (Appendix A) and DIV 4 scheme (Figure 2C). We repeatedly observed significant reductions with Usp7 and Usp10 siRNA, however effects with Usp9x, Usp21, and Senp3 siRNA did not continually meet significance.

#### 2.2.3. Validated siRNA Hits Lead to Target Knockdown in CTX from rTg4510 Mice

Next, we investigated the level of target knockdown archived with siRNAs for Usp7, Usp9x, Usp10, Usp21, and Senp3. Smartpool siRNA was applied at DIV 1 and target mRNA expression levels were measured at DIV 8 and DIV 15. Approximately 85% knockdown of MAPT, 75% knockdown of Usp9x, and 70% knockdown of Usp7, Usp10, Usp21 and Senp3 was achieved at both harvesting time points, indicating sufficient silencing of all five DUBs with their respective smartpool siRNA (Figure 2D).

#### 2.2.4. Usp7 and Usp10 Knockdown Does Not Reduce Human TAU Levels in CTX from rTg4510 Mice

To investigate whether the reduced seeded TAU aggregation achieved by the silencing of these five DUBs was mediated by the lowering of human TAU expression, we treated CTX with smartpool siRNA at DIV 1 and measured human TAU expression at DIV 15 by ELISA (Figure 2E). We achieved approximately 90% reduction in human TAU expression by siMAPT. We observed 41%, 34%, and 52% reduction in human TAU levels by knockdown of Usp9x, Usp21, and Senp3, respectively. Interestingly, knockdown of Usp7 and Usp10 did not affect soluble human TAU levels while significantly reducing seeded TAU aggregation, suggesting that silencing Usp7 and Usp10 can modulate TAU aggregation without reducing human TAU expression.

### 2.3. Knockdown Efficiency of Usp7 and Usp10 Correlates with Reduction in Seeded TAU Aggregation in CTX from rTg4510

We focused on Usp7 and Usp10 as silencing reduced seeded TAU aggregation most robustly without lowering TAU expression levels in rTg4510 CTX and deconvoluted their respective smartpool siRNA to identify the most efficient individual siRNA sequences. To this end, we tested the smartpool and the four individual siRNA sequences for Usp7 and Usp10 in parallel on efficiency to knockdown and to reduce seeded TAU aggregation (Figure 3) and additionally on effects on human TAU expression (Appendix A). Knockdown efficiency was in line with the efficiency to reduce seeded TAU aggregation. We confirmed approximately 70% knockdown with the smartpool siRNA Usp7 (Figure 3A) and smartpool siRNA Usp10 (Figure 3B) corresponding to significant 31–38% and 34–36% reduction in S1p- and P3-induced seeded TAU aggregation, respectively (Figure 3C,D). Individual siRNA Usp7 #15 was on par with smartpool siRNA Usp7, while siRNA Usp7 #14 achieved 54% knockdown resulting in only 23% reduction in S1p- and P3-induced seeded TAU aggregation (Figure 3A,C). Individual siRNAs Usp7 #13 and Usp7 #16 slightly outperformed smartpool siRNA Usp7 by reaching 73–75% knockdown and significant 39–48% and 38–43% reduction in S1p- and P3-induced seeded TAU aggregation, respectively (Figure 3A,C). Individual siRNA Usp10 #15 and #16 were on par with smartpool siRNA Usp10 on efficiency to knockdown and reduce seeded TAU aggregation, while siRNA Usp10 #14 outperformed smartpool siRNA Usp10 reaching 75% knockdown and significant 39–48% reduction in S1p- and P3-induced seeded TAU aggregation (Figure 3B,D). Individual siRNA Usp10 #13 led only to approximately 25% knockdown of Usp10 (Figure 3B) and did not reduce seeded TAU aggregation (Figure 3D). As expected, silencing Usp7 or Usp10 with their different siRNA sequences did not lower human TAU expression (Appendix A). Individual siRNA Usp7 #13 and Usp10 #14 were found most effective and selected for subsequent investigations.

### 2.4. Usp7 and Usp10 Knockdown Reduced Seeded Tau Aggregation Induced with TAU Seeds Derived from AD Brains in Wildtype CTX

Next, we investigated whether silencing of Usp7 and Usp10 reduced seeded Tau aggregation in a model without overexpression of mutant TAU by using cortical cultures isolated from wildtype mice. Here, endogenous murine Tau was templated by pathological TAU seeds from the sarkosyl-insoluble fraction isolated from AD brains (AD-TAU). We established that 1 µg/µL AD-TAU seeds added at DIV 7 were sufficient to induce seeded Tau aggregation which was measured by a mouse Tau aggregation immunoassay at DIV 15 (Figure 4). Seeded Tau aggregation did not result in impaired viability measured by resazurin assay and was dependent on endogenous mouse Tau expression as approximately 90% knockdown of mouse Tau with smartpool Mapt siRNA (siMapt) prevented seeded Tau aggregation by 91% (Figure 4A). Silencing of Usp7 and Usp10 with their respective siRNA Usp7 #13 and Usp10 #14 at DIV 1 led to significant 44% and 42% reduction, respectively, of AD-TAU-induced seeded Tau aggregation in wildtype CTX (Figure 4A), indicating that Usp7 and Usp10 knockdown was equally effective in reducing seeded Tau aggregation in cortical neurons expressing mouse Tau harboring no mutations.

### 2.5. USP7 and USP10 Inhibitors Reduced Seeded TAU Aggregation in rTg4510 and Wildtype Neurons

After identifying that knockdown of Usp7 and Usp10 reduced seeded TAU aggregation, we inhibited Usp7 and Usp10 pharmacologically with two specific USP7 inhibitors, FT671 [44] and GNE-6640 [45] and the USP10/13 inhibitor Spautin-1 [46] to investigate effects on seeded TAU aggregation in CTX from both rTg4510 and wildtype mice. First, we determined the tolerated inhibitor concentrations over a 7-day incubation period in CTX. USP7 inhibitors did not impair cell viability up to 5 µM, while Spautin-1 induced toxicity in concentrations higher than 2 µM. Subsequentially, inhibitors were incubated in non-toxic concentrations with rTg4510 neurons at DIV 8, 24 h after S1p and P3 TAU seed incubation at DIV 7 and seeded TAU aggregation was measured with the human TAU aggregation immunoassay (Figure 5A) and by quantifying the total intensity of pS422-TAU-positive inclusions normalized to the number of healthy nuclei (Figure 5B) at DIV 15. Inhibitors reduced concentration-dependently S1p- and P3-induced seeded TAU aggregation. Significant 39% or 33% and 34% reduction in S1p- and P3-induced seeded TAU aggregation was obtained when Usp7 was inhibited with 2 µM FT671 or 5 µM GNE-6640 and Usp10 was inhibited with 2 µM Spautin-1, respectively (Figure 5A). IC_50_ values, determined in the TAU aggregation assay (Figure 5A) and by quantifying the total intensity of pS422-TAU-positive inclusions normalized to the number of healthy nuclei (Figure 5B), were in comparable concentrations of 400–440 nM and 400–540 nM for FT671, 680–730 nM and 780–840 nM for GNE-6640, and 560–650 nM and 600–690 nM for Spautin-1, respectively. Additionally, we tested FT671 and Spautin-1 on AD-TAU-induced seeded Tau aggregation in wildtype CTX, using the same incubation scheme, and measured with the mouse Tau aggregation immunoassay at DIV 15. The USP7 inhibitor FT671 and the USP10/13 inhibitor Spautin-1 reduced concentration-dependently AD-TAU-induced seeded Tau aggregation with IC_50_ values of 410 and 590 nM, respectively (Figure 4B). Significant 34% and 33% reduction in AD-TAU-induced seeded Tau aggregation were obtained with 2 µM FT671 and 2 µM Spautin-1, respectively (Figure 4B), similar to the reductions in CTX from rTg4510 (Figure 5A).

### 2.6. USP7 and USP10 Inhibitors Increase Polyubiquitination Levels of Seeded TAU Aggregates in Neurons from rTg4510 Mice

TAU aggregates positive for pS422-TAU were also positive for polyubiquitin detected with the anti-ubiquitin antibody P4D1 (Figure 1A). We therefore assessed the ability of the inhibitors to reduce seeded TAU aggregation measured by the total intensity of polyubiquitin-positive TAU inclusions normalized to healthy nuclei and found comparable IC_50_ values of 460–510 nM for FT671, 740–780 nM for GNE-6640, and 580–640 nM for Spautin-1 (Figure 5C). We also measured the relative amounts of pS422-TAU and polyubiquitin on each TAU aggregate. Notably, when analyzing the polyubiquitin levels on the remaining inclusions in the presence of USP7 and USP10/13 inhibitors at their highest concentrations, we observed an approximately 30% significant increase in P4D1-positive spot intensity per inclusion (Figure 5D). The anti-ubiquitin antibody P4D1 recognizes polyubiquitin chains irrespective of linkage type and thus serves as a marker of total polyubiquitination. In contrast, no increase in pS422-TAU spot intensity per inclusion was detected under these conditions (Figure 5D). These results indicate that inhibition of Usp7 and Usp10 leads to an overall increase in the ubiquitination of TAU inclusions in CTX from rTg4510 mice.

### 2.7. Knockdown and Inhibition of Usp7 and Usp10 Reduce Seeded TAU Aggregation in Organotypic Hippocampal Slice Cultures (OHSCs) from rTg4510 Mice

Lastly, we investigated effects of Usp7 and Usp10 silencing and inhibition on seeded TAU aggregation in an ex vivo model in organotypic hippocampal slice cultures (OHSCs) which offered the benefit of studying seeded TAU aggregation in a three-dimensional and anatomically intact environment. OHSCs from rTg4510 did not exhibit spontaneous TAU aggregation; only the addition of S1p and P3 TAU seeds induced the formation of AT8-positive and pS422-TAU-positive inclusions, mainly in the CA1 region, consistent with seeded aggregation of intracellular TAU (Figure 6A,C). We observed comparable results whether seeded TAU aggregation was induced by S1p or P3 and TAU inclusions were detected with the AT8 or pS422-TAU antibody. As expected, when human TAU was silenced with siMAPT by 80% (Appendix A), approximately 90% inhibition of S1p- and P3-induced seeded TAU aggregation was achieved (Figure 6A,B). Knockdown efficiency of siRNA Usp7 #13 and Usp10 #14 in OHSCs from rTg4510 mice was approximately 65% (Appendix A). To determine effects of Usp7 and Usp10 knockdown on seeded TAU aggregation, siRNA Usp7 #13 and Usp10 #14 was applied to the slices at DIV 0, S1p and P3 TAU seeds were added at DIV 3, and seeded TAU aggregation detected as AT8-positive and pS422-TAU-positive inclusions at DIV 8 (Figure 6A). Significant 37% reductions in S1p- and P3-induced seeded TAU aggregation, detected as AT8-positive inclusions, were observed upon silencing of Usp7 and Usp10 (Figure 6A,B). Next, we investigated effects of Usp7 and Usp10 inhibition on seeded TAU aggregation in OHSCs, using 2 µM FT671, 5 µM GNE-6640 and 2 µM Spautin-1. Inhibitors were applied to the slices 2 h before S1p and P3 TAU seeds were added at DIV 3 and seeded TAU aggregation was detected as AT8-positive and pS422-TAU-positive inclusions at DIV 8. We observed significant 42% or 36% and 35% reduction in S1p- and P3-induced seeded TAU aggregation detected as AT8-positive inclusions when Usp7 was inhibited with FT671 or GNE-6640 and Usp10 was inhibited with Spautin-1, respectively (Figure 6C,D). Staining for total TAU or neurofilament as viability marker revealed healthy slices exhibiting no toxicity due to seeding, siRNA, or inhibitor treatment. Collectively, our results show that genetic and pharmacological inhibition of Usp7 and Usp10 significantly reduced seeded TAU aggregation in CTX and OHSCs.

## 3. Discussion

In this study, we identified Usp7 and Usp10 as two deubiquitinating enzymes (DUBs) whose downregulation or pharmacological inhibition robustly reduced seeded TAU aggregation in both cortical (CTX) and organotypic hippocampal slice cultures (OHSCs) derived from rTg4510 mice, as well as in wildtype CTX seeded with Alzheimer’s disease (AD)-derived pathological TAU. In these neuronal in vitro and ex vivo tauopathy models, seeded TAU aggregation manifested as prominent intracellular inclusions of hyperphosphorylated TAU within neurites and cell bodies. These inclusions were polyubiquitinated, thereby recapitulating key molecular features of pathological TAU observed in human tauopathies. Notably, the comparable efficacy of Usp7 and Usp10 knockdown or inhibition in wildtype CTX expressing only endogenous murine Tau suggests that inclusions composed of mouse Tau and human mutant P301L TAU are similarly targeted for deubiquitination by these enzymes.

In our DUB knockdown screen using CTX from rTg4510 mice, we identified and validated Usp7, Usp9x, Usp10, Usp21, and Senp3—all members of the cysteine-dependent protease superfamily—as significant siRNA hits. Among these, silencing of Usp7 and Usp10 consistently and significantly reduced seeded TAU aggregation, whereas knockdown of Usp9x, Usp21, and Senp3 produced more variable effects that did not always reach statistical significance. Importantly, while knockdown of Usp7 and Usp10 reduced seeded TAU aggregation without altering soluble human TAU levels, silencing of Usp9x, Usp21, and Senp3 led to a parallel reduction in both human TAU expression and seeded TAU aggregation. These results suggest that the apparent reduction in seeded TAU aggregation following Usp9x, Usp21, or Senp3 knockdown is likely secondary to decreased TAU expression rather than a direct effect on TAU aggregation. Consistent with this interpretation, previous work has shown that USP9 knockdown reduces TAU expression in cells and zebrafish [35], whereas no prior studies have linked USP21 or SENP3 to TAU regulation. In addition to USP7 [34], USP9 [35], and USP10 [26], several other deubiquitinating enzymes—including OTUB1 [32], UCH-L1 [33], USP11 [27], USP13 [28], and USP14 [36,37]—have been reported to deubiquitinate TAU. We initially hypothesized that these DUBs might also emerge as hits in our screen; however, siRNAs targeting Otub1, Uch-l1, and Usp14 did not affect seeded TAU aggregation in our primary screens. Moreover, although Usp11 and Usp13 siRNAs were among the initial 32 primary hits that modulated seeded tau aggregation, these effects were not reproduced in confirmation screens.

Our findings align with recent studies reporting Usp7 and USP10 as key regulators of TAU homeostasis. Usp7 has been shown to stabilize TAU through deubiquitination, and its silencing in a tauopathy mouse model ameliorated memory deficits, microglial activation, and TAU seeding activity [34]. Similarly, USP10 was reported to deubiquitinate TAU, with its overexpression increasing total and phosphorylated TAU levels and promoting TAU aggregation in neurons [26]. In addition, USP10 has been identified as a critical mediator of TAU recruitment to stress granules, thereby facilitating aggregation under stress conditions [30,47]. In our in vitro tauopathy model using rTg4510 cortical neurons knockdown of Usp7 and Usp10 significantly reduced seeded TAU aggregation without affecting normal human TAU levels. This observation suggests that their inhibition exerts a protective effect by enhancing the clearance of pathological TAU rather than by suppressing TAU expression. Given the essential role of TAU in maintaining microtubule stability and axonal transport, our findings highlight USP7 and USP10 as promising therapeutic targets to mitigate pathological TAU accumulation while preserving physiological TAU function.

Pharmacological inhibition of Usp7 and Usp10 using the selective inhibitors FT671, GNE-6640, and Spautin-1 effectively reduced seeded TAU aggregation, mirroring the effects observed following target silencing in CTX and OHSCs. Although Spautin-1 has been reported to inhibit both USP10 and USP13, our data indicate that only Usp10 inhibition is functionally relevant in this context, as siRNA-mediated knockdown of Usp13 did not impact seeded TAU aggregation. The IC_50_ values determined for inhibition of seeded TAU aggregation in CTX were consistent with previously reported potencies for these compounds—approximately 300 nM for FT671 [44], 750 nM for GNE-6640 [45], and 600–700 nM for Spautin-1 [46]. Furthermore, assessment of seeded TAU aggregation by immunocytochemistry, based on pS422-TAU-positive inclusions, yielded comparable IC_50_ values, supporting the robustness of our results across different assays. Our findings confirm that TAU inclusions are polyubiquitinated as indicated by P4D1 immunoreactivity. Consequently, inhibition of seeded TAU aggregation based on the measurement of polyubiquitin-positive inclusions delivered comparable IC_50_ values. Although P4D1 does not discriminate between specific polyubiquitin chain linkages, it reliably reflects the overall extent of polyubiquitination. Interestingly, inhibition of Usp7 and Usp10 led to a pronounced increase in the polyubiquitin intensity on the remaining TAU inclusions, particularly at the highest inhibitor concentrations. This enhanced ubiquitination likely facilitates the clearance of pathological TAU species, aligning with previous reports identifying TAU as a substrate for Usp7 [34] and USP10 [26].

## 4. Materials and Methods

### 4.1. Materials

All reagents were purchased from Sigma-Aldrich (St. Louis, MO, USA), unless otherwise stated. All antibodies used for immunocytochemistry and Western blot analysis are listed in the Appendix A. The USP7 inhibitors GNE-6640 (HY-112937) and FT671 (HY-107985) and the USP10/13 inhibitor Spautin-1 (HY-12990) were purchased from MedChemExpress EU (Sollentuna, Sweden). All Accell smartpool and individual siRNAs were purchased from Dharmacon (Horizon Discovery, Cambridge, UK).

### 4.2. Preparations of Pathological TAU Seeds from rTg4510 Mouse and AD Brains

The rTg4510 transgenic mice overexpress human 4R0N mutant P301L TAU in addition to endogenous murine TAU largely restricted to the forebrain by the CaMKIIα promoter [48]. Ten snap-frozen brains from 32 to 40 weeks old rTg4510 mice with advanced TAU pathology in the forebrain were pooled to generate a single batch of TAU seeds for the screening and validation studies. Forebrain homogenates were separated by differential centrifugation into Tris-buffered saline (TBS)-extractable (S1), S1 precipitate (S1p), and sarkosyl-insoluble pellet (P3) fractions as established previously [39]. To prepare TAU seeds from soluble S1p and insoluble P3 fractions, protease and phosphatase inhibitors and ion chelators in the buffers were omitted. Briefly, forebrains were weighted, homogenized in 10× volume homogenization buffer (50 mM Tris, 274 mM NaCl, 5 mM KCl, pH 8) of the forebrains weight and forebrain homogenates were separated by centrifugation at 27,000× *g* for 20 min at 4 °C in S1. The pellet was re-homogenized in 5× volume high salt/sucrose buffer (10 mM Tris, 800 mM NaCl, 10% sucrose, pH 7.4) of the forebrains weight and centrifuged as above. The supernatant was collected and incubated with 1% sarkosyl for 1 h at 37 °C, followed by centrifugation at 150,000× *g* for 1 h at 4 °C to obtain P3, dissolved in 0.5× volume Tris buffer (10 mM, pH 8) of the forebrain weight. S1 was separated by further centrifugation at 150,000× *g* for 20 min at 4 °C to obtain S1p, dissolved in 0.2× homogenization buffer of the S1 volume. The S1p and P3 batches were aliquoted to avoid freeze–thaw cycles of the TAU seeds and aliquots were stored at −80 °C. Animal experiments were performed in accordance with the European Communities Council Directive no. 86/609, the directives of the Danish National Committee on Animal Research Ethics, and Danish legislation on experimental animals (license no. 2014-15-0201-00339). Frozen postmortem samples of frontal cortices from four AD cases with Braak stage V/VI were obtained from Tissue Solutions Ltd. (Glasgow, UK) and pooled to generate a single AD-TAU seed batch for the validation studies. Sarkosyl-insoluble TAU (AD-TAU) seeds were extracted by 1% sarkosyl incubation (1 h, room temperature) and purified by several rounds of differential ultracentrifugation and sonication from the pooled gray matter homogenates as established previously [11]. The AD-TAU seed batch was aliquoted and stored at −80 °C. TAU concentrations in the different fractions were determined with the Innotest human total TAU ELISA (Fujirebio, Tokyo, Japan) following the manufacturer’s instructions. TAU seed batches were quality controlled by Western blot analysis and homogeneous time-resolved fluorescence (HTRF) human TAU aggregation immunoassay (see below). Additionally, the efficacy of the TAU seeds in inducing seeded tau aggregation in cortical cultures (see below) was measured and benchmarked to previous TAU seed batches.

### 4.3. Seeded TAU Aggregation, siRNA and Inhibitor Treatments in Cortical Cultures (CTX)

Murine cortical neurons (CTX) were isolated from day E14-16 rTg4510 or wildtype mouse embryos as described previously [49]. The generation of rTg4510 mice and the primer pairs to genotype rTg4510 embryos were described previously [48]. Briefly, single transgenic CaMKIIα promoter tTA activator mice were time-mated with single transgenic human mutant TAU responder mice. Pregnant females were euthanized and embryos were genotyped with brain DNA using the primer pair’s 5′-AGGCTGCTCTACACCTAGCT-3′ and 5′-CAGCGCATTAGAGCTGCTTA-3′ for the *tTA activator* transgene and 5′-CCAACGCCACCAGGATTC-3′ and 5′-AGCTGGGTGGTGTCTTTGGA-3′ for the human mutant *TAU* transgene while isolated embryonic cortices were kept in Hibernate E without calcium chloride (BrainBits LLC, Springfield, IL, USA) at 4 °C. Cortices from rTg4510 mouse embryos were selected and dissociated neurons were plated on 100 µg/mL poly-L-lysine coated dishes at a density of 0.13 × 10^6^ cells/cm^2^ (420,000 cells/mL, 100 µL/well, 96-well plate) and cultured in Neurobasal plus medium supplemented with 2% B-27 plus with antioxidants, 0.5 mM L-glutamine, 100 U/mL penicillin, and 0.1 mg/mL streptomycin, all solutions from Gibco-BRL Invitrogen (ThermoFisher Scientific, Waltham, MA, USA). At 4 days in vitro (DIV) half of the medium was replaced with fresh Neurobasal plus medium and 1 µM cytosine arabinoside added to halt proliferating cells. Thereafter, half of the medium was replaced with fresh medium every third day. The proportion of glia cells in the cultures was less than 10%, as assessed by an antibody against glia-fibrillary-acidic protein (GFAP) at DIV 8.

Accell smartpool and individual siRNAs were added at 1 µM final concentration according to the manufacturer’s instructions (Horizon Discovery, Cambridge, UK) directly in the Neurobasal plus medium at DIV 1, DIV 4, or DIV 6 for the DUB screen. At DIV 7, rTg4510 CTX were treated with 0.2 ng S1p or P3 TAU seeds or unseeded and incubated with a similar volume of PBS. Wildtype CTX were treated with 0.1 µg AD-TAU seeds or with a similar volume of PBS. At DIV 8, a complete medium change was performed to remove any residual TAU seeds not taken up and at DIV 12, cultures were fed with fresh Neurobasal plus medium. After the medium change at DIV 8, neurons were treated with USP inhibitors or similar volume of DMSO and at DIV 15, neurons were harvested to analyze seeded TAU aggregation by sequential TAU fractionation followed by Western blotting, HTRF TAU aggregation immunoassays, and TAU immunocytochemistry.

### 4.4. Seeded TAU Aggregation in Cortical Cultures Detected by Sequential TAU Fractionation and Western Blotting

Cell fractionations and Western blotting were performed as described previously [50]. Briefly, seeded neurons were lysed in Triton lysis buffer (1% Triton X-100 in 50 mM Tris, 150 mM NaCl, pH 7.6) supplemented with 1% protease inhibitor mixture (Roche, Basel, Switzerland), 1% phosphatase inhibitor cocktail I and II, and 0.2% benzonase and incubated on ice for 15 min. Following sonication, lysates were centrifuged at 100,000× *g* for 30 min at 4 °C and supernatants were kept as soluble Triton fractions. Pellets were washed in Triton lysis buffer and resuspended in SDS lysis buffer (1% SDS in 50 mM Tris, 150 mM NaCl, pH 7.6) containing 1% protease and phosphatase inhibitors. After sonication, SDS lysates were centrifuged at 100,000× *g* for 30 min at room temperature and supernatants were kept as insoluble SDS fractions. Protein concentrations from the resulting fractions were measured using the bicinchoninic acid (BCA) Protein Assay Kit from Pierce as per manufacturer’s instructions (ThermoFisher Scientific, Waltham, MA, USA). For detection of TAU, 10 µg total protein was dissolved in sodium dodecyl sulfate (SDS)-sample buffer containing dithiothreitol (DTT, 100 mM) and used for Western blotting as described previously [40]. TAU was detected using the rabbit polyclonal E1 antibody and the mouse monoclonal pS396-tau antibody D1.2 [41]. The E1 antibody was raised against amino acids 19–33 (GLGDRKDQGGYTMHQ) of the longest isoform of human TAU 2N4R [51] and recognizes exclusively human TAU at non-phosphorylated and phosphorylated epitopes. The pS396-tau antibody recognizes the phosphorylated epitope S396 of human TAU and murine Tau. All antibodies used for Western blot analysis are listed in the Appendix A.

### 4.5. Measurement of Seeded TAU Aggregation in CTX by HTRF TAU Aggregation Immunoassays

The HTRF TAU aggregation immunoassays (Cisbio, Codolet, France) are based on the principle of time-resolved measurement (TR) of fluorescence resonance energy transfer (FRET) between two fluorophores in close proximity coupled to a pair of antibodies recognizing either human TAU or mouse Tau. Using the same monoclonal antibody for the donor and acceptor fluorophores ensures that a TR-FRET signal is only obtained when the epitope is in a multimer form, thus quantifying selectively TAU aggregates. Cultures were lysed per well in 50 µL ice-cold lysis buffer (1% Triton X-100 in 50 mM Tris, 150 mM NaCl, pH 7.6) supplemented with 1% protease inhibitor mixture (Roche, Basel, Switzerland), 1% phosphatase inhibitor cocktail I and II, and 0.2% benzonase under shaking at 200 rpm for 45 min at 4 °C. Lysates from rTg4510 CTX were analyzed in the Cisbio human TAU aggregation kit (#6FTAUPEG) and lysates from wildtype CTX were analyzed in the Cisbio custom-made mouse Tau aggregation kit (#63ADK000E). Optimal sample dilutions were established for both TAU and Tau aggregation kits independently and according to the manufacturer’s protocols. In brief, rTg4510 lysates were diluted 4-fold in lysis buffer and 9 µL sample was mixed with 4.5 µL terbium (Tb)-cryptate donor fluorophore and 4.5 µL d2 acceptor fluorophore coupled human TAU antibody. Wildtype lysates were used undiluted and 13 µL sample was mixed with 2.5 µL Tb-cryptate donor fluorophore and 2.5 µL d2 acceptor fluorophore coupled mouse Tau antibody as described by the manufacturer. After 20 h incubation, HTRF signal was measured on a PHEARstar (BMG LABTECH, Ortenberg, Germany) using a pulsed 337 nm laser excitation and time delayed simultaneous dual emission at 665/620 nm. The data are presented as percent TAU aggregation and percent Tau aggregation above background normalized to protein concentrations (μg/μL) measured in the bicinchoninic acid (BCA) Protein Assay Kit from Pierce (ThermoFisher Scientific, Waltham, MA, USA) according to the manufacturer’s instructions. For detection of cell viability, the percentage of viable cells in the neuronal cultures was quantified at DIV 15 either before cell lysis with the resazurin assay kit (Abcam, Cambridge, UK) per manufacturer’s instructions or from parallel plates by their capacity to reduce 3-(4,5-dimethylthiazol-2-yl)-2,5-diphenyltetrazoliumbromide (MTT) after incubation with 0.5 mg/mL MTT for 60 min. Each condition was measured in at least 5 replicates (*n* = 5 wells) per culture plate.

### 4.6. Measurement of Seeded TAU Aggregation and Polyubiquitination in CTX by Immunocytochemistry

All antibodies used for immunocytochemistry are listed in the Appendix A. For immunocytochemistry, cells were fixed with ice-cold *v*/*v* 80% methanol for 10 min and washed in PBS. Blocking was performed in 1% BSA for 1 h, and cultures were subsequently incubated with the primary TAU antibodies, the mouse monoclonal AT8 (1:500, MN1020, Invitrogen, ThermoFisher Scientific, Waltham, MA, USA) and the rabbit polyclonal pS422-TAU antibody (1:500, 44764G, Invitrogen, ThermoFisher Scientific, Waltham, MA, USA) overnight at 4 °C. Alternatively, AT8 was combined with rabbit polyclonal DAKO total TAU antibody (1:5000, A0024, Agilent, Santa Clara, CA, USA) and pS422-TAU was combined with mouse monoclonal neurofilament, light chain (NEFL) antibody (1:200, 13-0400, Invitrogen, ThermoFisher Scientific, Waltham, MA, USA) to access network viability of the cultures. Ubiquitin was detected with the mouse monoclonal FK2 (1:500, 04-263, Merck Millipore, Darmstadt, Germany) or mouse monoclonal P4D1 antibody (1:500, sc8017, Santa Cruz Biotechnology, Dallas, TX, USA) used in combination with the pS422-TAU antibody. The anti-ubiquitin antibodies FK2 and P4D1 were raised against human polyubiquitinylated lysozyme and bovine polyubiquitin, respectively, and detect mono- and preferentially polyubiquitin in methanol-fixed specimens. After primary antibody staining, cultures were washed in PBS, and incubated with secondary anti-mouse (1:1000, A21202, Alexa488-donkey-anti-mouse, Invitrogen, ThermoFisher Scientific, Waltham, MA, USA) and anti-rabbit (1:1000, 715-545-150 Cy3-donkey-anti-rabbit, Jackson ImmunoResearch Europe Ltd., Ely, UK) antibodies and Hoechst 33342 (1:500, ThermoFisher Scientific, Waltham, MA, USA) for 1 h at room temperature. After two washing steps in PBS, cultures were subjected to high-content imaging with a Cellomics ArrayScan VTI HCS Reader (ThermoFisher Scientific, Waltham, MA, USA) equipped with a 20×/0.70 objective (Olympus, Tokyo, Japan) and analyzed in ThermoScientific HCS Studio: Cellomics Scan Version 6.6.0 (ThermoFisher Scientific, Waltham, MA, USA). Each condition was measured in 5 replicates (5 = wells) per culture plate, with each well imaged in 16 fields and an algorithm (Thermo Scientific HCS Studio: Cellomics Scan Version 6.6.3. Assay template: SpotDetector.v4) was created to analyze number, size, and intensity of pS422-TAU-positive and P4D1-polyubiquitin-positive inclusions. The spot parameters (number, size and intensity) were in close agreement and spot intensity selected as readout. Seeded TAU aggregation was measured as pS422-TAU-positive Spot Total Intensity/healthy nuclei and P4D1-positive Spot Total Intensity/healthy nuclei. Healthy nuclei were quantified using a different algorithm counting only Hoechst positive nuclei based on nuclear size (non-pyknotic) and fluorescent intensity from cells that were alive before fixation, and counts were used for the normalization of pS422-TAU and P4D1-polyubiquitin intensity per healthy nuclei. Additionally, the intensity levels of pS422-TAU and P4D1-polyubiquitin on each inclusion were determined using the SpotDetector algorithm to quantify pS422-TAU Spot Total Intensity/Spot and P4D1-polyubiquitin Spot Total Intensity/Spot.

### 4.7. Measurements of TAU Protein by ELISA

CTX from rTg4510 and wildtype were treated with 1 µM Accell target and control siRNA at DIV 1 or DIV 4 and harvested in cell extraction buffer (#FNN0011) from Invitrogen (ThermoFisher Scientific, Waltham, MA, USA) supplemented with 1 mM phenylmethylsulfonyl fluoride (PMSF) and 1% protease inhibitor mixture (Roche) at DIV 7 and DIV 15 to measure soluble human TAU and murine Tau levels using human TAU (Total) ELISA Kit (KHB0041) and mouse Tau (Total) ELISA Kit (KMB7011), respectively, from Invitrogen (ThermoFisher Scientific, Waltham, MA, USA) and to determine total protein concentration using the bicinchoninic acid (BCA) Protein Assay Kit from Pierce (ThermoFisher Scientific, Waltham, MA, USA) as per manufacturers’ instructions. Each condition was run in triplicates (*n* = 3 wells) measured in 3 technical replicates per experiment.

### 4.8. Seeded TAU Aggregation in Organotypic Hippocampal Slice Cultures (OHSCs)

Organotypic hippocampal slice cultures (OHSCs) were prepared based on a method described previously [52] and from postnatal day 6–7 rTg4510 mouse pups. Genotyping of pups was performed the day before preparation of OHSCs using the primer pairs described under cortical cultures. Hippocampi were isolated by dissection, and 350 μm thick hippocampal slices were sectioned with a tissue chopper (McIlwain, Hemmant, Australia) and transferred to a Petri dish containing cold Gey’s balanced salt solution with 6.5 mg/L glucose. The separation of the slices was performed with ultra-thin spatulas, and intact slices were transferred onto membrane inserts (PICM0RG50, Merck Millipore, Darmstadt, Germany) floating on top of 1.1 mL/well OHSC plating media (25% heat-inactivated horse serum, 50% OptiMEM with GlutaMAX, 25 mM glucose, 1 mM GlutaMAX, 2.5 mg/L phenol red, 25% HBSS with calcium and magnesium, 100 U/mL penicillin, and 0.1 mg/mL streptomycin). For knockdown experiments, OHSCs were treated with 1 µM Accell siRNA and cultured according to the manufacturer’s instructions in Accell delivery medium (Horizon Discovery, Cambridge, UK) supplemented with 2% B27 plus with antioxidants, 1 mM GlutaMAX, 25 mM glucose, 100 U/mL penicillin, and 0.1 mg/mL streptomycin at day 0 in vitro (DIV 0). OHSCs were maintained at 37 °C, 5% CO_2_, and 95% relative humidity. OHSC plating or Accell delivery medium was changed on DIV 3 to OHSC maintenance medium (Neurobasal plus medium, 2% B27 plus with antioxidants, 1 mM GlutaMAX, 25 mM glucose, 100 U/mL penicillin, and 0.1 mg/mL streptomycin) and seeding was performed by adding 0.2 ng S1p or P3 TAU seeds on top of each slice. All solutions were from Gibco-BRL Invitrogen (ThermoFisher Scientific, Waltham, MA, USA). Inhibitors or DMSO were added to the slices 2 h before seed addition in OHSC maintenance medium. OHSC maintenance medium was changed at DIV 6 and OHSCs were harvested at DIV 8 to investigate seeded TAU aggregation by immunocytochemistry.

### 4.9. Measurement of Seeded TAU Aggregation in OHSCs by TAU Immunocytochemistry

All antibodies used for immunocytochemistry are listed in the Appendix A. OHSCs were fixed in ice-cold 80% *v*/*v* methanol for 15 min and washed in PBS. Membrane inserts with OHSCs were stored at 4 °C in PBS containing 0.02% sodium azide until staining. OHSCs were permeabilized overnight in 0.5% TritonX-100 in Tris-buffered saline (TBS) and blocked for 6 h at room temperature in 10% bovine serum albumin (BSA) in PBS. OHSCs were subsequently soaked in primary mouse AT8 (1:500, MN1020, Invitrogen, ThermoFisher Scientific, Waltham, MA, USA) and rabbit pS422-TAU (1:500, 44764G, Invitrogen, ThermoFisher Scientific, Waltham, MA, USA) antibody solution overnight. Alternatively, AT8 was combined with rabbit DAKO total TAU antibody (1:5000, A0024, Agilent, Santa Clara, CA, USA) and pS422-TAU was combined with mouse neurofilament, light chain (NEFL) antibody (1:200, 13-0400, Invitrogen, ThermoFisher Scientific, Waltham, MA, USA). The primary antibody solution was removed by three sequential washing steps with 0.3% TritonX-100 in TBS. The secondary anti-mouse (1:1000, A21202, Alexa488-donkey-anti-mouse, Invitrogen, ThermoFisher Scientific, Waltham, MA, USA) and anti-rabbit (1:1000, 711-165-152, Cy3-donkey-anti-rabbit, Jackson ImmunoResearch Europe Ltd., Ely, UK) antibodies and Hoechst 33342 (1:500, ThermoFisher Scientific, Waltham, MA, USA) were applied for 3 h, followed by another 3 washes to remove the excess antibody. OHSCs were cut out of their membrane inserts and mounted on glass slides, shielded with coverslips, which were sealed with nail polish. Confocal images were acquired with a Leica DMi 8 microscope equipped with a 5×/0.15 objective (HC PL Fluotar, Leica, Wetzlar, Germany). Images were subsequently imported to Imaris (version 9.0.6, Oxford Instruments, Abingdon, UK), where a mask was created for the quantification of AT8-positive and pS422-TAU-positive inclusions (Imaris surface analysis with a smoothening value of 1 μm and pixel intensity threshold of 50). OHSC specimen with abnormal morphology or staining artifacts were excluded from the analysis. Each condition was measured in 3–5 slices per experiment.

### 4.10. Measurements of Gene Expression by Quantitative PCR

For gene expression quantification, the Fast SYBR™ Green Cells-to-CT™ Kit (#4402957) from Invitrogen (ThermoFisher Scientific, Waltham, MA, USA) was used to produce DNase I digested lysates from CTX and OHSC and by reverse-transcription reactions perform cDNA synthesis according to the manufacturer’s instructions. Quantitative PCR (qPCR) was performed using the SsoFAST SYBR EvaGreen PCR Master Mix (BIO-RAD, Hercules, CA, USA) according to the manufacturer’s instructions with primer pairs for the transgene human *TAU* (MAPT) 5′-CCAACGCCACCAGGATTC-3′ and 5′-AGCTGGGTGGTGTCTTTGGA-3′ from Eurofins Genomics (Ebersberg, Germany), mouse *Tau* (Mapt) 5′-TGTGGAGGTAATCTGTGAACTTC-3′ and 5′-AGGCTATGGATCTGAGGTCAAT-3′, *Usp10* 5′-GACTCTGTGTAGCATACAGGCT-3′ and 5′-CCTCACTAGGTTCGATGACTTCA-3′, *Usp9x* 5′-CAGAGTAATGATGATATTGCCTGCC-3′ and 5′-CGATCAAAGCAAGACTGAATGAAGT-3′, *Usp21* 5′-TGGCTCCTTCCACATGATATCTG-3′ and 5′-GAAGCATGTATTTCCTAGATTTCGGA-3′ from Primerdesign Ltd. (Chandler’s Ford, UK) and All-in-One qPCR primers for *Usp7* (MQP023609) 5′-CTCCCAGACCATGGGGTTTC-3′ and 5′-ATCTAACATTGCAGGCCGCT-3′ and *Senp3* (MQP106473) 5′-CCATCAGGGCTGGAAAGGTT-3′ and 5′-CTGGGTGAAGCTGAATGGCT-3′ from GeneCopoeia (Rockville, MD, USA). Normalization was performed using the geometric mean of four stably expressed mouse housekeeper genes (*E20Rik*, *Gpi1*, *Snrpd3*, and *Tbp*). All qPCR reactions were performed from 3 wells per experiment with technical duplicates which Cq values were averaged. Fold-change values were calculated using the ΔΔCt method and the ratio of target siRNA to control siRNA treated cells was expressed.

### 4.11. Statistical Analysis

All data analyses were performed using GraphPad Prism version 9 (GraphPad Software, San Diego, CA, USA). Comparisons involving three or more groups were analyzed using Brown–Forsythe and Welch ANOVA, followed by Dunnett’s T3 post hoc test, which accounts for unequal variances and sample sizes. Multiple treatment groups were compared to a single control group using a 95% confidence interval, with *p*-values adjusted for multiple comparisons. For the DUB knockdown screens, three independent experimental sets were conducted, corresponding to siRNA additions at DIV 1, DIV 4, and DIV 6. Within each set, the effects of smartpool siRNAs were compared to non-targeting siRNA controls using analysis of variance followed by Dunnett’s T3 post hoc test, to identify siRNAs that significantly increased or decreased seeded TAU aggregation. All reported *p*-values refer to post hoc comparisons. Data are presented as mean ± SD, and statistical significance is indicated as follows: *p* < 0.05 (*), *p* < 0.01 (**), *p* < 0.001 (***), and *p* < 0.0001 (****). Sample sizes (*n*) for each experiment and specific statistical analyses are provided in the corresponding figure legends.

## 5. Conclusions

The deubiquitinating enzymes Usp7 and Usp10 strongly influenced seeded TAU aggregation across various in vitro and ex vivo tauopathy models. Both genetic silencing and pharmacological inhibition of Usp7 and Usp10 significantly reduced TAU aggregation without altering soluble TAU levels, and this reduction was accompanied by increased polyubiquitination of residual TAU inclusions in cortical neurons. These findings provide further preclinical support that targeting USP7 and USP10 could promote degradation of pathological TAU and may represent a promising therapeutic strategy to slow disease progression in tauopathies such as Alzheimer’s disease.

## Figures and Tables

**Figure 1 ijms-26-11062-f001:**
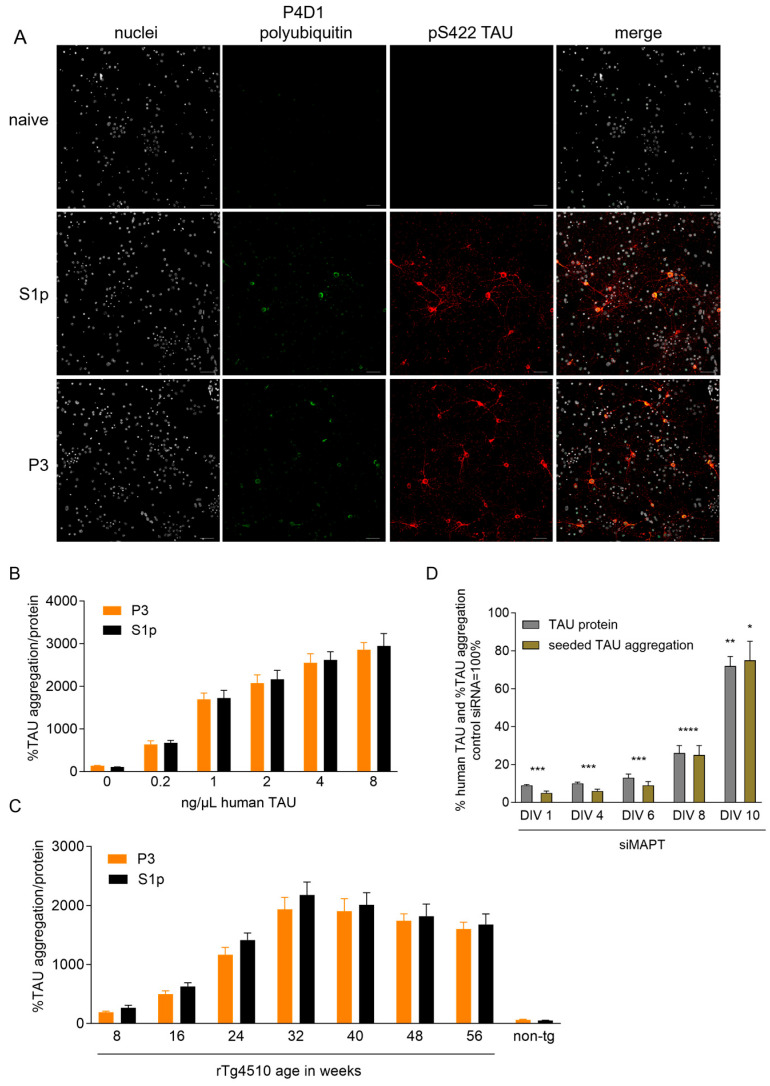
Defining optimal conditions for seeded TAU aggregation in CTX from rTg4510 mice. Cortical neurons isolated from rTg4510 mouse embryos were incubated at days in vitro (DIV) 7 with 2 ng/µL S1p or P3 TAU seeds isolated from brains of 32–40 weeks old rTg4510 mice or untreated (naive), methanol-fixed at DIV 15 and processed for immunocytochemistry. Hyperphosphorylated TAU was detected by pS422-TAU and polyubiquitination by P4D1 antibody. (**A**) Representative confocal images (20× objective) of nuclei stained with Hoechst (in gray), of polyubiquitination detected by P4D1 immunoreactivity (in green), and of seeded TAU aggregation detected by pS422 TAU immunoreactivity (in red). The scale bar corresponds to 50 µm. (**B**,**C**) Cortical neurons isolated from rTg4510 mouse embryos were incubated at DIV 7 with the indicated concentrations of S1p and P3 TAU seeds isolated from brains of 32–40 weeks old rTg4510 mice (**B**) and with S1p and P3 TAU seeds isolated from brains of rTg4510 mice of different ages (8, 16, 24, 32, 40, 48, and 56 weeks old) and of 56 weeks old non-transgenic (non-tg) mice (**C**) and harvested at DIV 15 to measure seeded TAU aggregation by the TAU aggregation assay. Data are presented as percentage seeded TAU aggregation normalized to total protein, mean ± SD from 5 wells representative from 3 independent experiments. (**D**) Cortical neurons isolated from rTg4510 mouse embryos were incubated at the indicated time points (DIV 1, DIV 4, DIV 6, DIV 8, DIV 10) with 1 µM siRNA MAPT (siMAPT) or non-targeting control and harvested at DIV 15 to measure human TAU protein levels by ELISA. Data are presented as bar graphs as means ± SD from 3 wells representative from 3 independent experiments in percentage with non-targeting control siRNA set to 100%. At DIV 7, CTX were treated with 2 ng/µL S1p TAU seeds isolated from brains of 32–40 weeks old rTg4510 mice and harvested at DIV 15 to measure seeded TAU aggregation by the TAU aggregation assay. Data are presented as bar graphs as means ± SD from 5 wells representative from 3 independent experiments in percentage with non-targeting control siRNA set to 100%. Brown–Forsythe and Welch ANOVA with Dunnett’s T3 post hoc test was performed; asterisks indicate significance (* *p* < 0.05, ** *p* < 0.01, *** *p* < 0.001, **** *p* < 0.0001).

**Figure 2 ijms-26-11062-f002:**
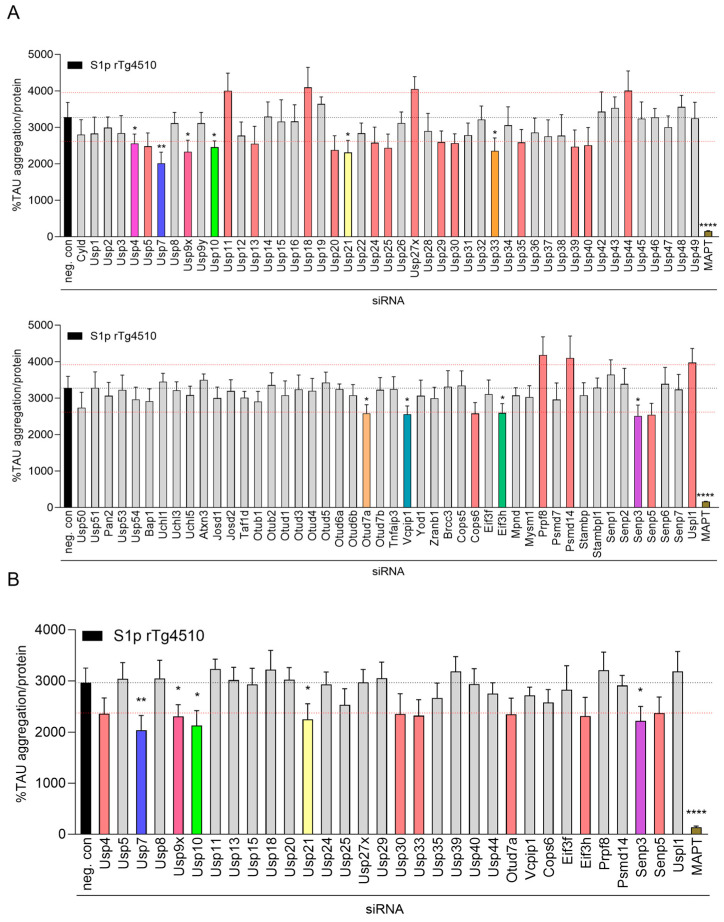
Silencing of Usp7 and Usp10 reduces seeded TAU aggregation without affecting human TAU levels in CTX from rTg4510 mice. Cortical neurons isolated from rTg4510 mouse embryos were incubated at DIV 4 with 1 µM smartpool siRNA targeting mouse DUBs, MAPT, or non-targeting control (neg. con). At DIV 7, cultures were treated with 2 ng/µL S1p TAU seeds isolated from brains of 32–40 weeks old rTg4510 mice and harvested at DIV 15 to measure seeded TAU aggregation by the TAU aggregation assay. Data are presented as percentage seeded TAU aggregation normalized to total protein, mean ± SD from 5 wells and mean ± SD from 10 wells for neg. con. The dotted black line indicates the negative control value, and the dotted red lines indicate the ±20% margins. (**A**) Primary screen with the 93 DUB siRNAs using 2 ng/µL S1p TAU seeds. (**B**) Confirmation screen with the 32 DUB siRNA hits using 2 ng/µL S1p TAU seeds. (**C**) Validation screen with 11 confirmed DUB siRNA hits using 2 ng/µL S1p or P3 TAU seeds. (**D**,**E**) Cortical neurons isolated from rTg4510 mouse embryos were incubated at DIV 1 with 1 µM smartpool siRNA targeting Usp7, Usp9x, Usp10, Usp21 and Senp3, MAPT, or non-targeting control (neg. con) and harvested to measure messenger RNA (mRNA) by qPCR at DIV 7 or DIV 15 (**D**) and human TAU protein by ELISA at DIV 15 (**E**). Data are presented as bar graphs as means ± SD from 3 wells representative from 2 independent experiments with non-targeting control siRNA set to 1 (**D**) and human TAU expression was normalized to total protein (**E**). Brown–Forsythe and Welch ANOVA with Dunnett’s T3 post hoc test was conducted; asterisks indicate significance (* *p* < 0.05, ** *p* < 0.01, *** *p* < 0.001, **** *p* < 0.0001).

**Figure 3 ijms-26-11062-f003:**
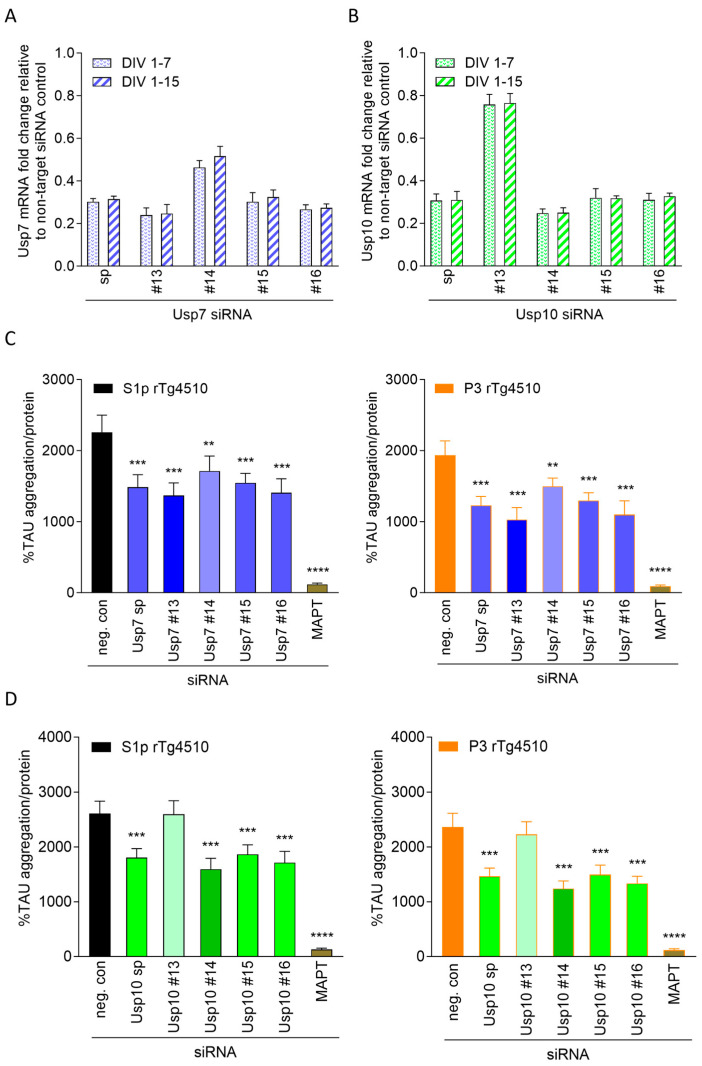
Usp7 and Usp10 knockdown efficiency determines reduction in seeded TAU aggregation in CTX from rTg4510 mice. Cortical neurons isolated from rTg4510 mouse embryos were incubated at DIV 1 with 1 µM smartpool (sp) or individual siRNAs (#13, #14, #15, and #16) targeting Usp7 (**A**) and Usp10 (**B**), MAPT, or non-targeting control (neg. con) and harvested to measure messenger RNA (mRNA) levels by qPCR at DIV 7 or DIV 15 (**A**,**B**). Data are presented as bar graphs as means ± SD from 3 wells representative from 3 independent experiments with non-targeting control siRNA set to 1. (**C**,**D**) At DIV 7, cultures were treated with 2 ng/µL S1p or P3 TAU seeds isolated from brains of 32–40 weeks old rTg4510 mice and harvested at DIV 15 to measure seeded TAU aggregation by the TAU aggregation assay. Data are presented as percentage seeded TAU aggregation normalized to total protein, mean ± SD from 5 wells representative from 3 independent experiments. Brown–Forsythe and Welch ANOVA with Dunnett’s T3 post hoc test was performed; asterisks indicate significance (** *p* < 0.01, *** *p* < 0.001, **** *p* < 0.0001).

**Figure 4 ijms-26-11062-f004:**
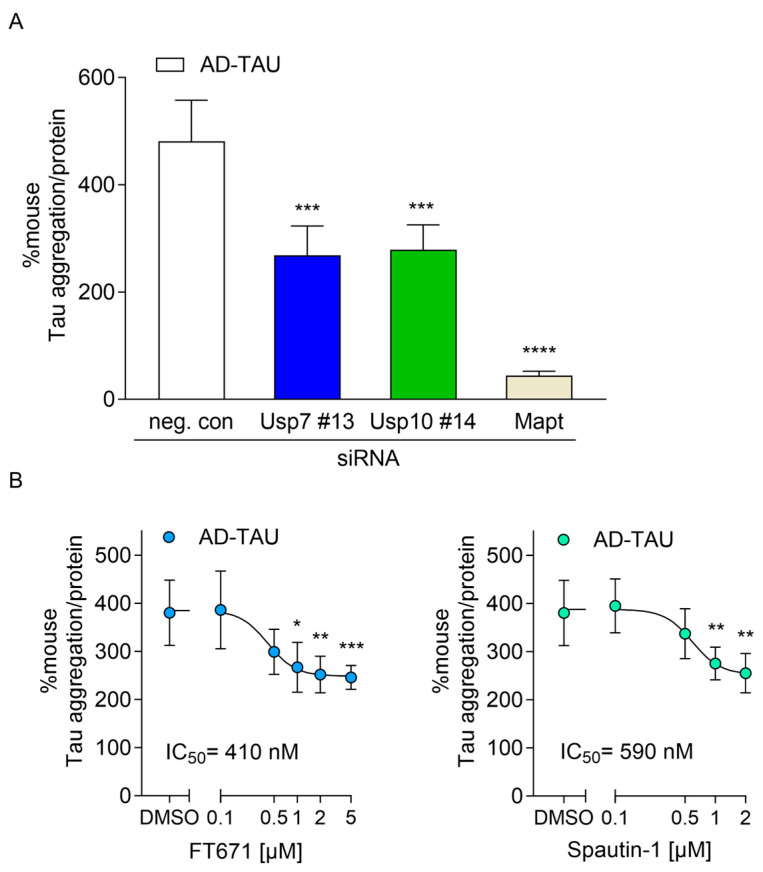
Silencing and inhibition of Usp7 and Usp10 reduces seeded Tau aggregation induced by AD-TAU seeds in wildtype CTX. (**A**) Cortical neurons isolated from wildtype mouse embryos were incubated at DIV 1 with 1 µM individual siRNA Usp7 #13, Usp10 #14, smartpool Mapt (siMapt) or non-targeting control (neg. con). At DIV 7, cultures were treated with 1 µg/µL AD-TAU seeds and harvested at DIV 15 to measure seeded Tau aggregation by the mouse Tau aggregation assay. (**B**) Cortical neurons isolated from wildtype mouse embryos were incubated at DIV 7 with 1 µg/µL AD-TAU seeds and at DIV 8 incubated with the indicated concentrations of FT671, Spautin-1, or DMSO and harvested at DIV 15 to measure seeded Tau aggregation by the mouse Tau aggregation assay. IC_50_ values of 410 nM for FT671 and of 590 nM for Spautin-1 were calculated. Data are presented as percentage seeded mouse Tau aggregation normalized to total protein, mean ± SD from 5 wells representative from 3 independent experiments. Brown–Forsythe and Welch ANOVA with Dunnett’s T3 post hoc test was performed; asterisks indicate significance (* *p* < 0.05, ** *p* < 0.01, *** *p* < 0.001, **** *p* < 0.0001).

**Figure 5 ijms-26-11062-f005:**
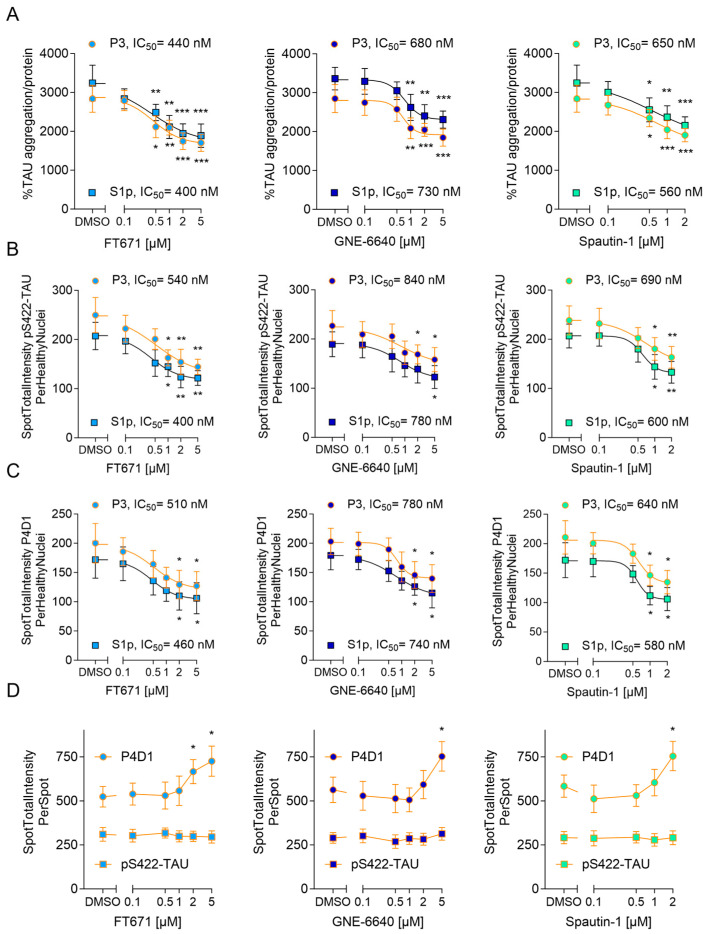
USP7 and USP10 inhibitors reduce seeded TAU aggregation and increase polyubiquitination of TAU inclusions in CTX from rTg4510 mice. Cortical neurons isolated from rTg4510 mouse embryos were incubated at DIV 7 with 2 ng/µL S1p or P3 TAU seeds and at DIV 8 incubated with the indicated concentrations of FT671, GNE-6044, Spautin-1 or DMSO and harvested at DIV 15 to measure seeded TAU aggregation by the TAU aggregation assay (**A**) presented as percentage seeded TAU aggregation normalized to total protein, mean ± SD from 5 wells, representative from 3 independent experiments and by immunocytochemistry and quantifications of pS422-TAU (**B**) and polyubiquitination by P4D1 (**C**) immunoreactivity shown as Spot Total Intensity/healthy nuclei count as mean ± SD from 5 wells representative from 3 independent experiments. Following IC_50_ values were calculated for FT671, GNE-6640, and Spautin-1, respectively: (**A**) 400–440 nM, 680–730 nM, and 560–650 nM; (**B**) 400–540 nM, 780–840 nM, and 600–690 nM; (**C**) 460–510 nM, 740–780 nM, and 580–640 nM. (**D**) P4D1 polyubiquitin and pS422-TAU immunoreactivity of P3-induced seeded TAU inclusions is presented as Spot Total Intensity/Spot as mean ± SD from 5 wells representative from 3 independent experiments. Brown–Forsythe and Welch ANOVA with Dunnett’s T3 post hoc test was conducted; asterisks indicate significance (* *p* < 0.05, ** *p* < 0.01, *** *p* < 0.001).

**Figure 6 ijms-26-11062-f006:**
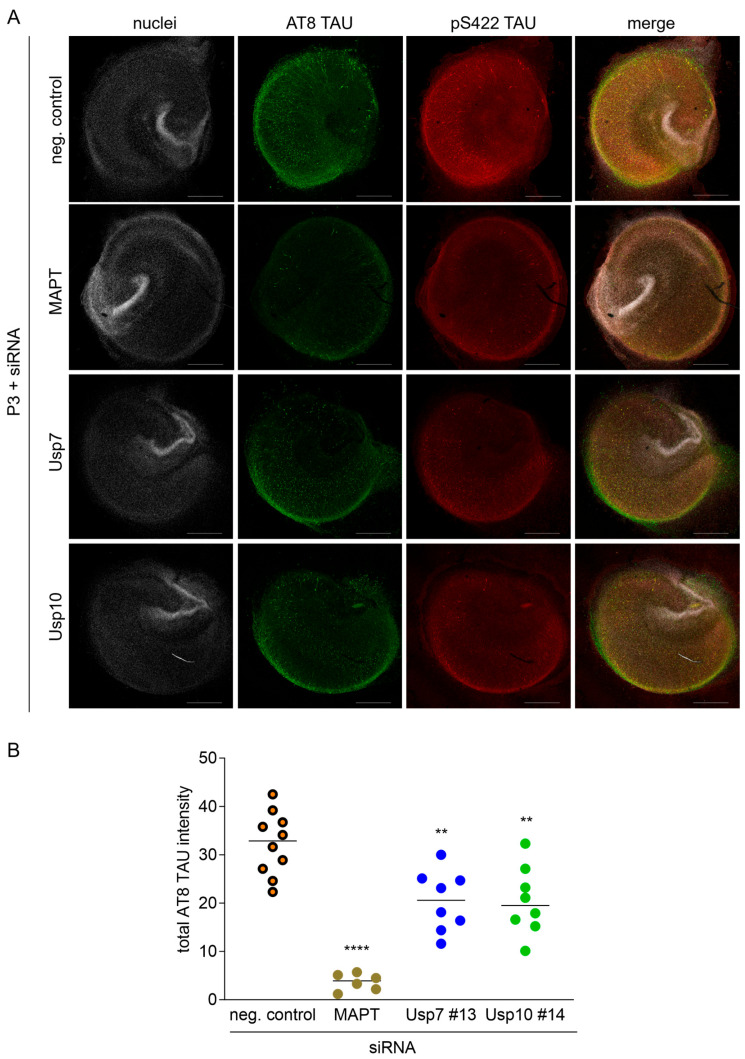
Silencing and inhibition of Usp7 and Usp10 reduces seeded TAU aggregation in OHSCs from rTg4510 mice. (**A**,**B**) Organotypic hippocampal slice cultures (OHSCs) isolated from rTg4510 mouse pups were treated at DIV 0 with 1 µM siRNA Usp7 #13 or Usp10 #14 or non-targeting control (neg. control). At DIV 3, slices were incubated with 2 ng/µL S1p or P3 TAU seeds per slice. (**C**,**D**) OHSCs isolated from rTg4510 mouse pups were incubated with the indicated concentrations of FT671, GNE-6044, Spautin-1 or DMSO 3 h before addition of 2 ng/µL S1p or P3 TAU seeds per slice at DIV 3. (**A**,**D**) Slices were methanol-fixed at DIV 8 and processed for AT8 and pS422-TAU immunocytochemistry to measure seeded TAU aggregation. Representative confocal images (5× objective) of nuclei stained with Hoechst (in gray) and P3-induced seeded TAU aggregation detected with AT8 (in green) and pS422-TAU (in red) immunoreactivity from 3 independent experiments. The scale bar corresponds to 500 µm (**A**,**C**). Quantifications of AT8 immunoreactivity from S1p and P3 seeded slices are shown as total AT8 intensity as mean ± SD from 6 to 10 individual slices presented as circles (**B**,**D**). Brown–Forsythe and Welch ANOVA with Dunnett’s T3 post hoc test was performed; asterisks indicate significance (** *p* < 0.01, *** *p* < 0.001, **** *p* < 0.0001).

## Data Availability

The original contributions presented in this study are included in the article/Appendix A. Further inquiries can be directed to the corresponding author.

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
