# Peer review of "Deubiquitinating Enzymes Ubiquitin-Specific Proteases 7 and 10 Regulate TAU Aggregation"

_ijms, 2025, doi:10.3390/ijms262211062_

Round 1
Reviewer 1 Report
Comments and Suggestions for Authors
In the manuscript authored by Christiane Volbracht and Karina Fog , the authors aimed to clarify how USP7 and USP10 regulate TAU aggregation across multiple experimental models. The work is well-conceived, technically solid, and provides valuable insights into DUB-mediated control of tauopathies. However, a few methodological aspects would benefit from clarification to strengthen reproducibility and interpretation.
-
Please provide more detail on buffer volumes used for S1p/P3 fractionation, storage conditions between experiments, and quality control measures to ensure consistency among different experiments.
- How the“Spot Total Intensity/viable cells” has been evaluated ? please add more details in the text.
-
In Figure 5D when authors quantifying the polyubiquitin signals on the remaining inclusions in the presence of USP7 and USP10/13 inhibitors, they determined an approx.
30% significant increase in P4D1 positive spot intensity. It is not clear if this data indicate longer chains or new ubiquitination sites? Authors should better discuss this in the text - To further improve the quality of the manuscript authors should improve figure clarity (scale bars, data visibility) and provide a summary table to include catalog/lot numbers/dilutions for all antibodies used.
Reviewer 2 Report
Comments and Suggestions for Authors
Some major comments:
Are there any network analysis or structure-based approaches that can be used to narrow down DUBs for inhibitions? Are there optimal combination of DUBS for inhibition?
Some sentences in the Results sections, such as "incubated CTX at DIV 7 with 2 ng/uL S1p or P3 TAU seeds isolated from brains of rTg4510
mice from different ages (8, 16, 24, 32, 40, 48, and 56 weeks old) and of 56 weeks old non-
transgenic (non-tg) mice as controls" should be moved to materials and Methods section. The Results section should just show the results - how to do the experiments should be moved to the Methods section.
Descriptions of figures should be provided ahead of figures. For example, only Figure 2A was cited and referred in the manuscript. Figures 2B to 2E should be cited and illustrated in the main text. Similar things happened to Figure 5.
The resolutions and explanation of Figure 2 need to be improved.
Information about some abbreviation is needed. For example, relationship between DIV and Usp in Figure 2.
Some figures from the supplmental should be moved back to the main text (as they were mentioned several times), such Supplementary Figure S 3-5
It is not clear how to select the 10 inhibitors. Any computationtal work?
Using Tau aggregation to as the indicator of inhibiting Ups7 may not be accurate. As the inhibitors may inhibit targets other Ups* to slow down the Tau aggregation.
Reviewer 3 Report
Comments and Suggestions for Authors
The manuscript by Volbracht et al. (ID ijms-3933622) presents findings of significant scientific relevance, outlining for the first time the contribution of Usp7 and Usp10 to the pathological accumulation of the TAU protein through the modulation of ubiquitin-dependent degradation pathways. Through an extensive siRNA screening targeting 93 deubiquitinating enzymes in rTg4510 cortical cultures, the authors demonstrate that gene silencing or pharmacological inhibition of Usp7 and Usp10 leads to a marked reduction in seeded TAU aggregation without affecting soluble TAU levels, accompanied by an increase in the polyubiquitination of residual inclusions.
Overall, the study provides an original and valuable contribution to the understanding of the molecular mechanisms regulating TAU homeostasis, highlighting the potential therapeutic relevance of USP7 and USP10 targeting as an innovative strategy for the treatment of Alzheimer’s disease and related tauopathies.
The manuscript addresses a topic of clear scientific interest and demonstrates a strong commitment to data collection and analysis. However, while the study shows good potential, I believe that several substantial revisions are needed to improve its clarity, readability, and overall methodological robustness.
A first issue concerns the quality of the figures, which currently appears limited. The figures are dense and lose definition when zoomed in, making the data difficult to interpret. Considering that visual representation is a key component of scientific communication, I recommend increasing the resolution of all images to maintain sharpness even at high magnifications.
Closely related to this point, the color scheme of the figures should also be reconsidered. The current black-and-white presentation makes it difficult to distinguish experimental groups and limits visual clarity. I suggest introducing the use of color, adopting harmonized and well-contrasted palettes, preferably accessible to color-blind readers. When properly calibrated, the use of color would not only improve the visual appeal of the manuscript but also enhance data readability and contribute to more effective communication of the results.
Another important observation concerns the statistical analysis section, which currently appears too concise. It is not sufficient to state that statistical information is reported below each graph; the reader should clearly understand, within the Materials and Methods section, the analytical approach employed. I therefore recommend providing a more detailed description of the statistical tests used, the significance thresholds (p < 0.05, including any corrections for multiple comparisons), and the analytical rationale, specifying independent and dependent variables and the reasoning behind the choice of tests. This would improve the transparency and reproducibility of the study and strengthen its methodological soundness.
In summary, I consider the manuscript to be scientifically valid and potentially impactful; however, it requires a thorough revision focusing on graphical and methodological aspects. Enhancing figure quality, adopting a more effective use of color, and expanding the description of statistical analyses would make the study much clearer, more accessible, and more consistent with the standards of international scientific publications.
Author Response
please see the attachment, no 3. reviewer

Round 2
Reviewer 3 Report
Comments and Suggestions for Authors
You have satisfactorily addressed all the requested revisions. Your article can therefore be accepted in its current form and is ready for publication